# High and Low Levels of an *NTRK2*-Driven Genetic Profile Affect Motor- and Cognition-Associated Frontal Gray Matter in Prodromal Huntington’s Disease

**DOI:** 10.3390/brainsci8070116

**Published:** 2018-06-22

**Authors:** Jennifer A. Ciarochi, Jingyu Liu, Vince Calhoun, Hans Johnson, Maria Misiura, H. Jeremy Bockholt, Flor A. Espinoza, Arvind Caprihan, Sergey Plis, Jessica A. Turner, Jane S. Paulsen

**Affiliations:** 1Neuroscience Institute, Georgia State University, Atlanta, GA 30302, USA; jciarochi1@student.gsu.edu; 2The Mind Research Network, Albuquerque, NM 87106, USA; jliu@mrn.org (J.L.); vcalhoun@mrn.org (V.C.); jbockholt@mrn.org (H.J.B.); fespinoza@mrn.org (F.A.E.); acaprihan@mrn.org (A.C.); splis@mrn.org (S.P.); 3Department of Electrical and Computer Engineering, University of New Mexico, Albuquerque, NM 87131, USA; 4Iowa Mental Health Clinical Research Center, Department of Psychiatry, University of Iowa, Iowa City, IA 52242, USA; hans-johnson@uiowa.edu (H.J.); jane-paulsen@uiowa.edu (J.S.P.); 5Department of Psychology, Georgia State University, Atlanta, GA 30302, USA; mmisiura1@student.gsu.edu; 6Department of Neurology, University of Iowa, Iowa City, IA 52242, USA; 7Department of Psychology, University of Iowa, Iowa City, IA 52242, USA

**Keywords:** Huntington’s disease, brain-derived neurotrophic factor, tropomyosin receptor kinase B, supplementary motor, independent component analysis

## Abstract

This study assessed how *BDNF* (brain-derived neurotrophic factor) and other genes involved in its signaling influence brain structure and clinical functioning in pre-diagnosis Huntington’s disease (HD). Parallel independent component analysis (pICA), a multivariate method for identifying correlated patterns in multimodal datasets, was applied to gray matter concentration (GMC) and genomic data from a sizeable PREDICT-HD prodromal cohort (*N* = 715). pICA identified a genetic component highlighting *NTRK2*, which encodes BDNF’s TrkB receptor, that correlated with a GMC component including supplementary motor, precentral/premotor cortex, and other frontal areas (*p* < 0.001); this association appeared to be driven by participants with high or low levels of the genetic profile. The frontal GMC profile correlated with cognitive and motor variables (Trail Making Test A (*p* = 0.03); Stroop Color (*p* = 0.017); Stroop Interference (*p* = 0.04); Symbol Digit Modalities Test (*p* = 0.031); Total Motor Score (*p* = 0.01)). A top-weighted *NTRK2* variant (rs2277193) was protectively associated with Trail Making Test B (*p* = 0.007); greater minor allele numbers were linked to a better performance. These results support the idea of a protective role of *NTRK2* in prodromal HD, particularly in individuals with certain genotypes, and suggest that this gene may influence the preservation of frontal gray matter that is important for clinical functioning.

## 1. Introduction

### 1.1. Huntington’s Disease

Huntington’s disease (HD) is a progressive, heritable condition characterized by chorea (involuntary motion) as well as cognitive alterations spanning executive functioning, working memory, olfactory and facial recognition, and emotional processing [1]. HD, along with Alzheimer’s disease (AD) and Parkinson’s disease (PD), is a proteinopathy distinguished by regionally-selective neuronal death and protein misfolding that manifests as expanded huntingtin in HD, Lewy bodies in PD, and β-amyloid plaques in AD [2]. Unfortunately, across these conditions limited treatment options and no known cures are available. However, their shared features have sparked speculation about common underlying mechanisms, and the delayed-onset of these disorders raises the appealing possibility of developing treatments that postpone onset indefinitely, effectively eradicating the disease.

A promising way to identify treatment targets is to characterize the earliest changes before the onset of HD. Motor impairments associated with HD, such as dystonia and chorea, often lead to diagnosis because their disruptiveness prompts affected individuals to seek medical attention. However, cognitive symptoms and alterations in brain volume and morphology are already present more than a decade before diagnosis, during a period known as the prodrome [3]. In keeping with this, PREDICT-HD is a multi-site research study aiming to identify the earliest changes in the HD prodrome, with the hopes of identifying targets for the earliest possible interventions [3]. PREDICT-HD has amassed a comprehensive dataset of genomic, structural and functional magnetic resonance imaging (MRI and fMRI), diffusion tensor imaging (DTI), cognitive and motor assays, cytosine-adenine-guanine (CAG)-repeat information, and demographic variables from over 1449 prodromal HD and control participants, including longitudinal data from over 900 participants.

Although there is no cure for HD, its cause is known. An abnormally large cytosine-adenine-guanine (CAG) expansion (≥36 repeats) at an *HTT* exon 1 locus determines future HD development. The *HTT* gene encodes huntingtin protein, which is widely expressed in the brain and central nervous system [4]. Abnormally-expanded *HTT* encodes mutant huntingtin (mHTT), which compromises numerous cellular processes including endocytosis and secretion, calcium homeostasis [5], glutamatergic synaptic functioning [6], vesicular transport [7], mitochondrial functioning [8], p53 signaling [7], apoptosis, and transcription [9].

### 1.2. Effects of Multiple Genes and Variants

Both within and outside the realm of huntingtin’s interactions, several lines of evidence implicate non-*HTT* factors as modulators of prodromal progression and HD onset. Moreover, the reduced genetic complexity of HD makes it tractable to disentangle onset-protection and susceptibility factors. *HTT* CAG-expansion length considerably influences age at diagnosis and can be used to estimate the age of, or time to, HD onset. Despite strong prediction accuracy for many prodromal individuals, some outcomes deviate from expectations. For example, one PREDICT-HD participant with 44 CAG-repeats lacked positive diagnosis at the age of 71 years, and 13 participants with <41 repeats reached the age of 70 years with no diagnosis. HD onset prediction (based on age and CAG-repeat number) is most accurate in individuals with >44 repeats and increasingly variable as the repeat number decreases, and different disease progression rates are often observed in persons with the same number of CAG-repeats. These examples highlight the onset variability and suggest that additional genetic factors may promote or suppress HD conversion (especially at lower CAG-repeat numbers), yet little is known about non-*HTT* genetic factors that account for variability in the rapidity and severity of HD symptoms and onset.

The influence of such factors is likely also reflected by differences in brain structure and clinical functioning throughout the prodrome. Known polygenetic neural effects suggest that multiple genes may modulate decline; this is in keeping with the prevailing common disease-common variant model, which posits that the combined effects of multiple common nucleotide variants, or single nucleotide polymorphisms (SNPs), with weak individual effects may confer disease susceptibility or resistance [10]. At an individual level, these polymorphisms may occur in several, sometimes interacting genes, bestowing weak enough effects to fall below statistical thresholds and avoid elimination via natural selection [10]. We observe similar covariance in the brain; even in disorders with regionally concentrated damage, multiple brain regions and cell types are usually affected.

### 1.3. Benefits of Multivariate Methods

Interactions among multiple genes can have complex effects on disease phenotypes [10]. Univariate methods such as genome-wide association studies (GWAS) have dominated large-scale human genetic studies, despite an inability to capture this important covariation [10,11]. Univariate tests require tens of thousands of participants, which can be impossible to achieve in rare clinical populations, and must correct for many statistical tests. These stringent statistical standards can result in an overshadowing of small-to-moderate genetic effects and obscured interpretation of impacted biological pathways, as results generally consist of a few of the most significant genes, each of which is involved in multiple cellular processes and pathways. For these reasons, multivariate methods may be more suitable for extensive genetic studies, especially in rare clinical populations with fewer available study participants. Rather than assessing related points, multivariate tests find interrelated patterns and can detect weak effects in high-dimensional data.

### 1.4. Parallel Independent Component Analysis (pICA)

Combined effects of nucleotide-level differences (or SNPs) on gray matter concentration (GMC) across the brain can be assessed across the genome and in candidate genes using the multivariate method of parallel ICA (pICA) [10,11]. Through the simultaneous analysis of multimodal data, pICA can isolate groups of correlated SNPs into novel, maximally-independent networks that affect patterns of GMC in a population. In other words, a person whose genome aligns with a pICA SNP profile that is correlated with a GMC profile will also likely display a brain structure consistent with that GMC profile. pICA has been successfully applied to other clinical populations, including schizophrenia and Alzheimer’s disease [12,13], which share key features with HD such as delayed onset, regional and cellular selectivity of atrophy, and cognitive abnormalities.

Like other multivariate methods, pICA is optimal for examining multi-gene interactions because it considers the cumulative effects of changes at multiple loci, likely accounting for more variation than the strongest changes in single genes [10]. Consequently, it requires fewer statistical tests than univariate Genome-Wide Scanning (GWS) analyses (<30 compared to hundreds of thousands). An additional advantage of pICA is that it permits both hypothesis-driven testing (via inclusion of reference SNPs) and the exploration of new and unexpected patterns (disease-related or otherwise) that connect genes and their expression to brain structure and function. These advantages allow affected pathways and gene networks to be more thoroughly defined.

### 1.5. Brain-Derived Neurotrophic Factor (BDNF)-Signaling Genes (a Candidate Pathway)

Genes that interact with mHTT and are involved in mHTT-compromised processes are strong candidate HD-progression mediators. One such target is brain-derived neurotrophic factor (BDNF). BDNF mediates neurogenesis, and accumulating evidence suggests its importance in HD development and onset [9,14]. BDNF co-localizes with huntingtin in 99% of pyramidal motor cortical neurons and 75% of striatal neurons, and is necessary for healthy cortico-striatal synaptic activity and striatal GABA-ergic medium spiny neuron (MSN) survival [9].

mHTT interferes with BDNF transcription and vesicular transport (Figure 1). (a) Huntingtin enhances BDNF microtubule transport by binding to HAP1, which engages vesicle transport proteins. mHTT binds too tightly to HAP1, inhibiting transport. (b) Huntingtin stimulates transcription from the BDNF exon II promoter, which is 60% less active in cells overexpressing mHTT [9]. Like many neuronal-gene promoters, BDNF is regulated by repressor element 1/neuron-restrictive silencer element (RE1/NRSE), which is modulated by RE1 silencing transcription factor/neuron-restrictive silencer factor (REST/NRSF) [7]. REST is a neuron-specific, master gene repressor that binds to BDNF promoter II RE1 sites and recruits a co-repressor complex that includes Sin3A and REST co-repressor (coREST). Huntingtin indirectly sequesters REST in the cytoplasm by interacting with HAP1 and REST-interacting LIM domain protein (RILP), which directly binds REST/NRSF to mediate translocation to the nucleus. mHTT disrupts this complex by failing to isolate REST in the cytoplasm, leading to its increase in the nucleus and consequent recruitment of transcriptional repressors.

BDNF is particularly relevant to prodromal HD. Compared to controls, asymptomatic HD transgenic mice have reduced striatal BDNF that is lower at higher CAG-repeat numbers, indicating a prodromal BDNF deficit [14]. Further BDNF reduction in these mice lowers onset age and worsens motor symptoms, correlating with brain morphology changes. Furthermore, BDNF deficiency only modestly contributes to early-life MSN survival, yet significantly reduces MSNs (by 35%) in later life, consistent with delayed HD onset. Interestingly, the striatum does not produce its own BDNF, but receives ~95% from the cortex and the remainder from the substantia nigra pars compacta, amygdala, and thalamus [14]. Thus, prodromal BDNF suggests possible early deficits in regions that supply striatal BDNF. Given that BDNF mRNA and protein reductions are present in prodromal and diagnosed HD and are causally related to symptom severity and HD onset, factors that interact with BDNF likely confer additional symptom and onset variability.

### 1.6. The Present Study

Using parallel ICA with reference (pICAr), this study leveraged legacy imaging, genomic, and clinical data from PREDICT-HD to investigate how SNPs in BDNF-signaling genes impact clinical functioning and patterns of brain morphology in prodromal HD. As an expansion of the pICA results, we assessed the effects of four individual SNPs in *NTRK2* (which encodes BDNF’s TrkB receptor) on frontal gray matter and clinical functioning. Profiles including SNPs from multiple genes were anticipated to correlate with gray matter concentration profiles and clinical functioning with maximum effects in BDNF-related, HD-compromised pathways.

## 2. Materials and Methods

### 2.1. Participants

A cohort of 715 expansion-positive (≥36 CAG-repeats) prodromal PREDICT-HD participants were included in the study (448 females and 267 males; ages 18–82 years; mean CAG-repeat number = 42.5 (SD = 2.5)). Participant data included genotyping, T1-weighted structural MR images, cognitive and motor variables, and demographics (including age, sex, years of education, and CAG-repeat number) from PREDICT-HD [15]. Exclusion criteria included poor genomic or imaging data quality, relatedness to another participant, or the presence of any other central nervous system disorder or unstable medical or psychiatric condition. All PREDICT-HD participants provided written, informed consent and were treated in accordance with protocols approved by each participating institution’s internal review board. Participants underwent genotyping before study enrolment, and those with more than 35 CAG-repeats who did not meet criteria for HD diagnosis were designated as prodromal.

### 2.2. Data Availability

SNP data included in this study are publicly available from dbGAP (Study Accession: phs000222.v4.p2). Other PREDICT-HD data, including baseline T1-weighted MR images (used as input files in this study), subcortical and cortical segmentations, and longitudinal clinical measurements, are available on the public download site (ftp://ftp.ncbi.nlm.nih.gov/dbgap/studies/phs000222). Release of the specific genetic and imaging component data generated in this study is forthcoming; however, similar ICA-generated gray matter imaging components from a different PREDICT-HD study by the lead author have been made available in the latest study version (Study Accession: phs000222.v5.p2; Dataset Name: SBM_sMRI; Dataset Accession: pht006857.v1.p2). Access to the specific component data from the study may be requested from the corresponding author.

### 2.3. Cognitive and Motor Variables

Seven clinical variables were selected and tested for brain structural and genetic associations (outlined below), based on their established clinical reliability and sensitivity to prodromal HD progression [16]. Because we were interested in genetic and brain-structural effects on cognitive and motor functioning, we analyzed the portions of the Unified Huntington’s Disease Rating Scale (UHDRS) relevant to these domains. The UHDRS includes four sections measuring movement, cognition, behavior, and functional capacity. We assessed the Total Motor Score (UHDRS-TMS), which comprises the movement portion of the UHDRS and is a sum of the scores on the individual motor variables (oculo-motor function, dysarthria, chorea, dystonia, gait, and postural stability). We also analyzed the Symbol Digit Modalities Test (SDMT) and the Stroop test (color, word, and interference conditions), which are two of the three parts of the UHDRS cognition section (we did not assess verbal fluency). In addition to the UHDRS-TMS, SDMT, and Stroop Interference, we also analyzed Trail-Making Tests A and B (TMTA and TMTB), which are not part of the UHDRS. We chose these tests because of our specific interest in cognition and motor functioning, and because previous work by the PREDICT-HD group demonstrated that these measures are particularly sensitive to prodromal changes in brain structure [16].

Briefly, the SDMT measures working memory, complex scanning, and processing speed, and is an adaptation of the Wechsler Digit Symbol subtest [17,18]. Participants are provided with a key of symbols paired with numbers at the top of the test page. On the same test sheet, a series of numbers are presented in a horizontal row, and the task is to fill in the symbol matching each number in the sequence as quickly and accurately as possible. Raw scores represent the number of correctly completed items within 90 s; thus, higher scores indicate better performance [19].

The Stroop Color and Word Test consists of three 45-s conditions that measure basic attention and inhibition of an overlearned response [20,21]. For the color condition, the task is to identify colors presented on stimulus cards. For the word condition, the task is to read color names presented in black ink. Both the color and word conditions measure basic attention. For the interference condition, participants identify the ink color in which color-names are printed, rather than reading the color name itself. For example, for the word “blue” printed in green ink, the correct response is “green.” The interference condition measures the ability to inhibit the dominant (or automatic processing) response, which is to read the word. For each of the three conditions, raw scores reflect the number of correct trial responses, and higher scores thus reflect better performance.

TMTA and TMTB measure visual attention and task switching [22,23]. For TMTA, the task is to sequentially connect a series of numbered circles (e.g., 1-2-3-4, etc.) as quickly as possible. For TMTB, participants consecutively connect numbers and letters in ascending/alphabetical order, alternating between numbers and letters (e.g., 1-A-2-B-3-C, etc.). For both TMTA and TMTB, raw scores reflect the time (in seconds) taken to complete the task. Thus, higher scores reflect poorer performance.

The UHDRS-TMS assesses several indicators of motor performance spanning oculomotor function, bradykinesia, chorea, dystonia, gait, and postural stability [1,24]. Higher scores indicate poorer motor functioning.

### 2.4. Genomic Data Preprocessing

Data for 1,160,231 SNPs assayed with the Illumina Human 1M platform were downloaded from dbGAP (Study Accession: phs000222.v4.p2). Data were filtered at a 5% missingness rate per sample and per SNP, and SNPs with a minor allele frequency (MAF) of >0.05 were selected. After linkage disequilibrium (LD) pruning (*r*^2^ > 0.5), the full genomic dataset consisted of 305,271 SNPs that were included in further analyses. Only one participant per family was included in the study; the family structure was determined using PLINK identity-by-descent analysis, where *p* > 0.18 indicates relatedness (http://zzz.bwh.harvard.edu/plink/ibdibs.shtml). The top 10 multi-dimensional scaling (MDS) factors were used to correct for population structure.

### 2.5. Imaging Data Collection

High-resolution anatomical MR images were collected at 33 sites using General Electric, Phillips, and Siemens scanners with field strengths of 1.5 T (Tesla) or 3 T. The study used a standardized acquisition protocol that was modified for each site by our MR physicist. Secondary to upgrades and acquisition changes over the 12-year study, additional variation occurred. A total of 50 site/scanner field strength combinations were analyzed, with at least three participants from each different MRI scanner. Images at each site were obtained using three-dimensional (3D) T1-weighted inversion recovery turboflash (MP-RAGE) sequences. Then, 1.5 T scans were collected using General Electric and Siemens scanners. The Siemens protocol was constructed to be similar to the General Electric scan parameters: GRAPPA factor, 900 ms TI (inversion time), 2530 ms TR (relaxation time), 3.09 ms TE (excitation time), 256 mm × 256 mm field of view (FoV), 10° flip angle, 240 coronal slices with 1 mm slice thickness, 256 × 128 matrix with 1/4 phase FoV, 220 Hz/pixel receiver bandwidth. Protocol for 3 T scanners commonly involved a sagittal localizing series followed by acquisition of an axial three-dimensional (3D) volumetric spoiled gradient recalled acquisition in a steady-state (GRASS) sequence, using the following scan parameters: ~1 mm × 1 mm × 1.5 mm voxel size, 18 ms TR, 3 ms TE, 24 cm FoV, 20° flip angle, 124 slices with 1.5 mm slice thickness, 0 mm gap, 256 × 192 matrix with 3/4 phase FoV, number of excitations (NEX) = 2.

### 2.6. Imaging Data Preprocessing

Images were aligned with the anterior commissure-posterior commissure (AC–PC) plane, resampled with 1 mm isotropic voxels to correct for inhomogeneity [25], and preprocessed using SPM8 (http://www.fil.ion.ucl.ac.uk/spm/software/spm8/). Images were segmented into gray matter, unmodulated (to isolate GM concentration), and normalized to the same SPM8 Montreal Neurological Institute (MNI) template. Voxels were re-sliced to 2 × 2 × 2 mm^3^, and images were smoothed by a 10 × 10 × 10 mm^3^ full-width-half-maximum (FWHM) Gaussian kernel. Processed images were 90 × 109 × 91 voxels in size. A linear regression model was applied to each GM voxel to account for the effects of age, sex, and site (inclusive of field strength); the site variable was coded using 49 dummy variables.

### 2.7. Parallel ICA with Reference (pICAr)

Prodromal GMC and SNP patterns were identified and tested for correlations by applying pICAr to the GMC imaging and the genomic data. SNPs within nine genes implicated in BDNF-signaling were included as references: *BDNF*, *NGFR*, *NTRK2*, *RCOR1*, *SIN3A*, *SORT1*, *HAP1*, *REST*, and *RILP* (see Table 1). SNPs within 20 kbp of these genes were also included to capture regulatory elements in intronic and intergenic regions.

PICA is an extension of ICA, a robust and popular method for isolating maximally independent sources from a mixed signal [10,11,26]. In a general ICA model, X = AS: X is an observation (i.e., subject-by-variable matrix); S is a statistically independent component matrix (component-by-variables); and A is the loading coefficient or mixing matrix, the representation of each component in the subject or sample (subject-by-component). PICA performs this extraction simultaneously on two modalities, X1 and X2. For this experiment, X1 was a participants-by-voxels matrix of the masked GMC images, and X2 was a participants-by-loci matrix of the 305,271 SNPs. pICAr decomposes the observation into maximally independent sources (component matrices S1 and S2) by updating W1 and W2 (unmixing matrices) to optimize F_1_ (Infomax algorithm that maximizes the independence of modality-1 components), F_2_ (modified Infomax that optimizes the modality-2 component independence AND similarity to the reference matrix), and F_3_ (maximizes the correlations between the two modalities’ loading coefficient matrices, A1 and A2) [10,11]. Each reference is a vector containing alleles of a gene likely to be in linkage disequilibrium, meaning they co-occur more or less frequently than expected for independent loci. A reference vector for each gene listed in Table 1 comprises the reference matrix r. pICA outputs a loading coefficient for each participant for each SNP and imaging profile, representing how much the participant’s genome and brain structure matches each profile detected in the sample.(1)$$\begin{array}{l} X_{d}=A_{d}S_{d}\to S_{d}=W_{d}X_{d},A_{d}=W_{d}^{-1},d=1,2\\ Y_{d}=\frac{1}{1+e^{-U_{d}}},U_{d}=W_{d}X_{d}+W_{d0} \end{array}$$
(2)$$F_{1}=\max\left\{H\left(Y_{1}\right)\right\}=\max\left\{-E\left[\ln f_{y_{1}}\left(Y_{1}\right)\right]\right\}$$
(3)$$\begin{array}{l} F_{2}=\max\left\{\lambda H\left(Y_{2}\right)+\left(1-\lambda \right)\left[-dist^{2}\left(\tilde{r},\left|{\tilde{S}}_{2k}\right|\right)\right]\right\}\\ =\max\left\{\lambda \left(-E\left[\ln f_{y2}\left(Y_{2}\right)\right]\right)+\left(1-\lambda \right)\left(-{∥\left|W_{2k}{\tilde{X}}_{2}\right|-\tilde{r}∥}_{2}^{2}\right)\right\} \end{array}$$
(4)$$F_{3}=\max\left\{\sum_{i,j}Corr^{2}\left(A_{1i},A_{2i}\right)\right\}=\max\left\{\sum_{i,j}\frac{Cov^{2}\left(A_{1i},A_{2i}\right)}{Var\left(A_{1i}\right)Var\left(A_{2i}\right)}\right\}$$

### 2.8. SNP and GMC Correlations with Clinical Variables

We then queried SNPs of interest (i.e., SNPs within Table 1 BDNF-signaling genes that were also highlighted in the pICA results) for correlations with the cognitive and motor variables. Five SNPs in NTRK2 were tested in separate multivariate general linear model (GLM) tests using SPSS [27]; the clinical measures were dependent variables, and the SNP value (a continuous variable between 0 and 2) was a covariate. We similarly examined associations between the clinical variables and a pICA SNP and GMC component that were significantly correlated with each other. Here, the clinical measures were the dependent variables and the SNP or GMC loading coefficient was a covariate.

### 2.9. Associations of Top-Weighted Component SNPs with Clinical Variables

To test whether the observed effects were explained by the entire component cumulatively or were largely the effect of heavily-weighted SNPs, restricted genetic coefficients based on the most important SNPs in the component were calculated. First, the distribution of the weights for the 305,271 SNPs in the SNP component was fitted to a logistic distribution. Based on this, SNPs with weights more than 4.25 standard deviations (SDs) from the mean were selected as top contributing SNPs (*N* = 61). Weights for these 61 SNPs within the component (61 × 1 matrix) were multiplied by participant genotypes for these SNPs (715 × 61 matrix) to yield new participant loadings for the top 61 contributing SNPs. These new loadings were correlated with loadings for the full component at r(713) = 0.77 (two-tailed significance of *p* < 0.001). A list of the top 61 SNPs, associated genes, and weights in the component is available in Appendix A.

### 2.10. Confirmation of Significant Results

Permutation testing and leave-N-subjects-out (10-fold, 10% of total sample) cross-validation were used to verify significant pICA component correlations [10,28]. Permutation testing involves a random shuffling/mismatching of participant SNP and GMC data, which are then subjected to pICA and correlation testing. The permutation testing is performed multiple (in this case, 1000) times, allowing a null distribution to be formed that reveals the likelihood of obtaining the significant GMC-SNP correlation by chance. Leave-N-out (or 10-fold validation) is achieved by running pICA on 10 separate datasets, each containing 90% of the full data, to determine if the results from the original analysis are replicable.

### 2.11. Regression Influence Plot

An influence matrix was generated using the regression influence plot function in R [29,30] to pinpoint any individuals disproportionately driving the outcome of the significant GMC-SNP component correlation. This function outputs a plot of studentized residuals by hat values as well as a data frame containing hat values, studentized residuals, and Cook’s distances. The approach efficiently describes the influence that each dependent variable value has on each fitted or predicted value.

## 3. Results

### 3.1. pICAr

In our PREDICT-HD prodromal cohort, pICAr detected a significantly correlated GMC-SNP component pair (r(713) = 0.17, *p* < 0.001). This association remained significant after Bonferroni correction for multiple testing. The GMC component most strongly represented supplementary motor, superior, mid, and medial frontal, and precentral/primary motor regions (Figure 2). The correlated SNP component had strong contributions from the *NTRK2* gene, which encodes BDNF’s high-affinity receptor type (TrkB) (Figure 3). Four intronic *NTRK2* SNPs (rs11140810, rs4877289, rs10868241, rs2277193) were among the top 10 SNPs contributing to the component. Other top genes included *CDK14*, *FAM114A1*, and *HEATR4* (Table 2).

### 3.2. SNP and GMC Correlations with Clinical Variables

The frontal GMC component highlighted by pICA (and associated with the *NTRK2* SNP component) was significantly correlated with four of the seven queried clinical variables, including UHDRS total motor score (F(1,672) = 6.8, *p* = 0.009, surviving Bonferroni multiple testing correction), TMTA (F(1,672) = 6.3, *p* = 0.01, passing Bonferroni), Stroop Color (F(1,672) = 4.7, *p* = 0.03), and Stroop Interference (F(1,672) = 3.8, *p* = 0.05). This component also approached a significant correlation with the other three clinical variables (SDMT: F(1,672) = 3.5, *p* = 0.06; Stroop Total: F(1,672) = 3.4, *p* = 0.07; TMTB: F(1,672) = 3.1, *p* = 0.08)). In each case, a higher GMC was linked to improved performance.

The frontal-GMC-associated *NTRK2* SNP component was not significantly related to the clinical measures. However, one of the top *NTRK2* SNPs contributing to the component (rs2277193) was significantly associated with TMTB (F(1,672) = 6.7, *p* = 0.01), an effect that survived Bonferroni correction. Here, the minor allele was associated with better TMTB performance. This SNP also approached a significant correlation with Stroop Interference (F(1,672) = 3.6, *p* = 0.058), with the minor allele similarly being associated with better performance. *NTRK2* SNP rs111408010 was also correlated with UHDRS-TMS (F(1,672) = 3.7, *p* = 0.05), with a greater minor allele number being associated with better functioning (lower TMS score), but this effect did not withstand multiple testing correction.

### 3.3. Confirmation of Significant Results

The pICA results passed permutation testing; the ratio of correlation values above maximally linked components was 0.013, translating to about a 1% likelihood that the results were obtained by chance. However, the results did not replicate in all the 10-fold validation runs, likely because the GMC-SNP component correlation was driven by several individuals with high or low levels of the SNP component (Figure 4). The visual interpretation was confirmed by the regression influence analysis, which highlighted 25 participants that dominated the association (Appendix A). All of these participants had SNP component values 2–4 SDs above or below the mean.

### 3.4. Associations of Top-Weighted Component SNPs with Clinical Variables

Similar to tests with the full SNP component, clinical associations with weights comprised of the top 61 component SNPs yielded no significant results. Derived *p* and *r*^2^ values were slightly weaker for the top 61 weights but were essentially comparable.

## 4. Discussion

### 4.1. High or Low Levels of the NTRK2 SNP Profile Affect Prodromal Frontal GMC

Our results suggest that a strong or weak presence of the *NTRK2*-weighted genetic profile may be required to significantly affect the frontal gray matter profile. The genetic effect on frontal/supplementary motor GMC in individuals in the tail of the sample was still strong enough to be significant and apparent within the larger sample. In keeping with this, all 25 individuals driving the significant SNP-GMC component relationship (according to the regression influence analysis) had SNP component scores 2–4 SDs above or below the mean; only three individuals with SNP component values in this range were not highlighted by the regression influence test. By contrast, only six of these participants also had GMC component values ≥2 SDs from the mean in either direction, suggesting that the effect is specific to the SNP component. These participants did not stand out in any other obvious way from the full prodromal sample, and represented both sexes and a variety of ages, sites, and CAG-repeat numbers.

Four *NTRK2* SNPs were among the top 10 SNPs contributing to the GMC-related SNP component. *NTRK2* (Neurotrophic Receptor Tyrosine Kinase 2) is a large (358,613 base) chromosome 9 gene located between *SLC28A3* (solute carrier family 28, sodium-coupled nucleoside transporter member 3) and *AGTPBP1* (ATP/GTP binding protein 1). *NTRK2* is widely expressed in the brain (Figure 5) and encodes BDNF’s high-affinity TrkB (tropomyosin receptor kinase B) receptor. TrkB has important cellular functions that may underlie *NTRK2*’s prominence and protective role in this study.

TrkB binds with BDNF, neurotrophin-4 (NTF4), and neurotrophin-3 (NTF3), and regulates neuronal differentiation, growth, and survival as well as synaptic plasticity and the transcription of cell survival genes [31]. Thus, it affects both long- and short-term learning and memory by mediating short-term synaptic function as well as long-term potentiation. TrkB has also been implicated in apoptosis-suppression and the promotion of communication between neurons and glial cells.

Regarding BDNF, TrkB is preferentially activated by mature BDNF and expressed in indirect pathway striatal MSNs [9]. Following activity-dependent, anterograde transport by cortical afferents, BDNF can act post-synaptically on TrkB receptors to inhibit GABAergic or enhance glutamatergic synaptic transmission. By interacting with presynaptic TrkB receptors, BDNF can also be retrogradely transported to the cell body, where it stimulates cortical glutamate release. TrkB also responds to neuronal damage; its mRNA increases following excitotoxic lesions, and reduced TrkB is associated with neurodegeneration in Alzheimer’s disease. In the context of HD, mHTT reduces TrkB expression in a CAG-dependent manner (in R6/1 mice), an effect that is rescued by mHTT inactivation [9].

Like other neurotrophins, BDNF is synthesized in the endoplasmic reticulum (ER) as a large (32 kDa) precursor protein (pro-BDNF) [9], then translocated to the Golgi complex and secretory vesicles, proteolytically cleaved, and released as mature BDNF (14 kDa). P75, BDNF’s low-affinity receptor (encoded by NGFR), is preferentially activated by pro-BDNF and promotes apoptosis. There is evidence for a cortical pro-BDNF overabundance in HD, which is one explanation for the lack of consensus on cortical BDNF reductions in HD and the prodrome [9]; pro-BDNF increases could mask BDNF reductions when probed with methods that do not differentiate between pro- and mature BDNF. In humans, a highly conserved and fairly common (20–30% heterozygotes, ~4% homozygotes) BDNF polymorphism (Val66Met) in the 5′ region that encodes pro-BDNF is associated with memory deficits and multiple disorders [14]. Mature BDNF production is not affected by Val66Met, but pro-BDNF trafficking and packaging are substantially impacted. Interestingly, mHTT significantly blocks the post-Golgi trafficking of Val66Val (but not Val66Met) BDNF, although neither allele is associated with disrupted transport from the ER to the Golgi.

Considering the roles of BDNF’s high- and low-affinity receptor types in promoting cell survival and apoptosis, respectively, a pro- to mature BDNF ratio imbalance could contribute to the abnormal apoptosis observed in HD [14]. TrkB receptor underrepresentation could hinder BDNF’s ability to increase its own expression. Similarly, p75 receptor or pro-BDNF overabundance could amplify apoptosis and contribute to disease-related decline. The results of the present study suggest that *NTRK2* variations could influence this balance protectively, promoting cell survival and preserving frontal gray matter and cognitive and motor performance.

### 4.2. The SNP-GMC Component Correlation is Not an Aggregate Effect of the Entire SNP Component

Weights comprised of the top 61 SNPs contributing to the component (rather than the full component with weights for all 305,271 SNPs) yielded comparable results when evaluated with the GMC profile and clinical variables. Derived *p* and r^2^ values were only slightly weaker in analyses using the top 61 SNP weights, indicating that the original results do not reflect the aggregate effects of the entire SNP component but are rather primarily due to the top contributing SNPs.

### 4.3. The NTRK2-Associated Frontal Gray Matter Profile Is Related to Prodromal Cognitive and Motor Functioning

The correlated GMC component, containing supplementary motor, superior, middle, and medial frontal, and precentral/primary motor cortex, was significantly related to both cognitive (TMTA, Stroop Color, Stroop Interference) and motor (TMS) performance in this cohort. Although the variance in individual clinical performance measures accounted for by the GMC component was small, such effects are typical when comparing single clinical variables to brain structural measures; the observed associations were consistent with the literature reporting structural and functional changes in these regions that correlate with altered performance on these tasks in independent samples [32,33,34,35,36]. Although each of the cognitive tasks has a motor component (writing or verbalization is required for response), the stronger correlation for TMS compared to the cognitive measures likely reflects the strength of the supplementary and primary motor cortex in the GMC component and the increased demand on these regions for motor relative to cognitive functioning [37]. This may also explain why significant effects were not observed for SDMT and Stroop Word, as motor involvement in these tasks is relatively limited.

### 4.4. Top Contributing NTRK2 SNPs

Among the top 10 SNPs contributing to the frontally-related SNP component were rs11140810, rs4877289, rs10868241, and rs2277193; no other gene was represented more than once in the top 10 SNPs. Interestingly, three of these four SNPs are between *NTRK2* exons 19 and 20 (the location of an alternative stop codon), with rs2277193 (the TMTB-related SNP) being the closest to exon 19. In fact, rs2277193 is only 799 base pairs from rs11140810, the strongest *NTRK2* contributor to the component, and it is 3577 base pairs from the other top SNP in this region (rs10868241). The placement of these SNPs (as well as the fourth *NTRK2* SNP, located between exons 16 and 17) is fitting, as truncated *NTRK2* isoforms that lack the catalytic tyrosine kinase domain (generated by alternate terminal exon 16 or exon 19) are considered the most clinically relevant [38].

Of the *NTRK2* SNPs, rs11140810 had the strongest weight in the component. The Lieber Institute for Brain Development’s (LIBD) expression quantitative trait loci (eQTL) browser, comprised of dorsolateral prefrontal cortex (DLPFC) expression data from 237 healthy controls and 175 schizophrenia patients, identified this SNP as an eQTL for *NTRK2* in the DLPFC at the expressed region level [39]. According to Braineac (http://www.braineac.org), an eQTL database from 134 healthy human brains [40], the regulatory effects of rs11140810 on *NTRK2* are strongest in the occipital lobe, thalamus, and frontal cortex. Other top affected genes include *SLC28A3*, *HNRNPK*, *MIR7-1*, and *NAA35*. Table 3 lists the top component SNPs and their strongest reported regulatory effects on genes in the brain. HaploReg, a tool for visualizing 1000 Genomes Project SNP LD information in conjunction with Roadmap Epigenomics and ENCODE chromatin state and protein binding annotation, shows an association of rs11140810 with markers of H3K27 acetylation (linked to active transcription) in the brain (Table 4), as well as altered regulatory motifs including forkhead box (Foxa), glioma-associated oncogene (GLI), hypermethylated in cancer (Hic1), and zinc finger protein (Zec) [41].

The nearby rs2277193 had a protective effect on TMTB, in which the minor allele was associated with improved performance. In the brain, this SNP exhibits maximum regulatory effects on many of the same genes, albeit a less pronounced influence on *NTRK2* [40]. According to Haploreg, this SNP is associated with expression-enhancing chromatin states in the brain (Table 4), altered glucocorticoid receptor (GR) and paired-box protein (Pax-6) regulatory motifs, and *AGTPBP1* expression in the prefrontal cortex (PFC) [41].

rs4877289 is correlated with H3K4me1 (enhancer activity associated) in anterior caudate and cingulate gyrus, and is in strong LD with other *NTRK2* SNPs associated with histone enhancement and promotion, protein binding, altered regulatory motifs, and eQTL hits. For example, rs11140785 (LD = 0.94) bound to STAT3 in ChIP-Seq and is associated with altered myocyte enhancer factor (Mef2) regulatory motif and expression-enhancing chromatin changes in the hippocampus, substantia nigra, caudate, inferior temporal, cingulate gyrus, angular gyrus, and DLPFC. In the brain, this variant most strongly affects *RMI1*, especially in the frontal cortex [40].

Like rs11140810, rs10868241 is an identified eQTL for *NTRK2* in the DLPFC at the expressed region level in control and disease populations [39]. It is also associated with enhancer and promoter-associated active histone modifications in several brain regions (Table 4), and is in strong LD with SNPs linked to enhancer histone marks and altered regulatory motifs (for example, rs4457413 (LD *r*^2^ = 0.97) is associated with alteration of 12 regulatory motifs and with histone enhancers in the temporal lobe and PFC). In the brain, rs10868241 most strongly regulates many of the same genes as the other top *NTRK2* SNPs (*SLC28A3*, *HNRNPK*, *MIR7-1*, *NTRK2*, *AGTPBP1*, *RMI1*) [40], especially in the frontal cortex and hippocampus.

In summary, the four *NTRK2* SNPs that contributed most to the SNP component were situated near alternative stop codons (rs4877289 between exons 16 and 17, and the others between exons 19 and 20); these SNPs act on *NTRK2*, its neighboring *SLC28A3* and *AGTPBP1* genes, and other genes (*HNRNPK*, *MIR7-1*, *NAA35*, *RM1*) in and outside of the brain tissue. These genes are implicated in apoptosis regulation, smooth muscle cell proliferation, mRNA splicing, neurotransmission, translational control, SUMOylation, Golgi and ER retrograde transport, and the shortening of long polyglutamate chains, among other functions (Table 4). The effects of these *NTRK2* SNPs on genes are most common and pronounced in the frontal cortex (in keeping with the SNP component’s connection to frontal gray matter), followed by the thalamus, putamen, and cerebellum. Table 5 summarizes histone modifications and altered regulatory motifs associated with these SNPs.

It is important to consider LD pruning when interpreting these results. Although two of these SNPs (rs1114081 and rs2277193) are not in high LD with other SNPs, rs4877289 is in high LD with three SNPs (rs10780691, rs10868235, rs10868230; *r*^2^: 0.92–0.98) and rs10868241 is in high LD with nine SNPs (rs7858707, rs10116596, rs10122796, rs7030319, rs4329345, rs11464614, rs10780693, rs11140813, rs4457413; *r*^2^: 0.97–1.0). Thus, the effects of the latter two SNPs in this study may be attributable to an SNP in high LD that was pruned from the dataset.

### 4.5. Top Contributing SNPs Outside of NTRK2

Although *NTRK2* SNPs were disproportionally represented in the top component SNPs, other top SNPs included intronic *CDK14*, *FAM114A1*, and *HEATR4* SNPs, as well as intergenic SNPs (rs548321 closest to *LRRC55*, rs427790 between *MIR181A1* and *NR5A2*). Many of these genes are implicated in cancer regulation (Appendix A). Overall, compared to top component *NTRK2* SNPs, top SNPs outside of *NTRK2* were less commonly associated with histone modifications in the brain, but were more often reported to influence regulatory motifs and gene expression outside the brain. In Braineac data, the 10 strongest SNP influences on brain gene expression among our top 10 component SNPs were exhibited by three of the SNPs outside of *NTRK2* (rs7655305, rs8012614, rs548321); rs7655305 accounted for half of these effects. Appendix A, a complement to Table 5, summarizes brain histone modifications and altered regulatory motifs associated with these SNPs (the *HEATR4* SNP is omitted due to the absence of relevant findings). Of these SNPs, the intergenic SNPs rs548321 (*LRRC55*-associated) and rs427790 (*MIR181A1* and *NR5A2*-associated) share perhaps the most overlap with the *NTRK2* SNPs; rs2277193 (*NTRK2*) and rs548321 both alter GR (glucocorticoid receptor) and Pax (paired box) regulatory motifs (Pax-4 for rs548321 and Pax-6 for rs2277193) and are associated with histone changes in the brain. rs11140810 (*NTRK2*) and rs427790 alter the same Zec transcription factor and two distinct Fox regulatory motifs (Foxj1 for rs427790 and Foxa for rs11140810). Each of these SNPs are outlined in more detail below.

The *CDK14* variant (rs7801922) contributed to the SNP component more than any other SNP. This intronic SNP is 68,161 bp upstream of *CDK14* exon 1, and is an eQTL in the lungs (*GTPBP10*), tibial artery (*CDK14*), tibial nerve (*GTPBP10*), and thyroid (*DPY19L2P4*) [43]. In the brain, rs7801922 is an eQTL for the DLPFC in psychiatric populations [39]. In healthy populations, this variant most affects genes that exhibit reduced expression in white matter relative to other brain tissue types (e.g., *STEAP2*, *DPY19L2P4*, *C7orf63*) [40]. Appendix A lists examples of gene regulation related to this SNP. rs7801922 is associated with altered regulatory motifs (ecotropic virus integration site (Evi-1), GATA-binding factor (GATA), interferon regulatory factor (Irf), and suppressor of essential function (SEF-1)), and is in LD with three other *CDK14* SNPs associated with other regulatory motif alterations [41].

An intronic *HEATR4* (HEAT-repeat containing 4) variant was another top SNP. HEAT-repeats are amino acid domains that often have roles in protein-protein interactions, and are considered important for intracellular transport, microtubule dynamics, and chromosomal separation. Huntingtin is one of the first proteins in which HEAT-repeats were discovered (HEAT stands for Huntingtin, Elongation factor 3, protein phosphatase 2A, and TOR) [44]. The *HEATR4* SNP that was prominent in our genetic component (rs8012614) is an eQTL for *ACOT4* (exon, expressed region, gene, and transcript levels), *DNAL1* (expressed region level), and *HEATR4* (expressed region level) in the DLPFC [39]; in controls, the SNP was an eQTL for *ACOT4* only (exon, expressed region, and gene levels). Additionally, this SNP has more Genotype-Tissue Expression (GTEx) eQTL hits than any other top contributing SNP, although these are mostly outside the brain in adipose, muscle, skin, heart, pancreas, and thyroid tissue [43]. The effect of this SNP on *DNAL1* is particularly interesting, as this gene is not only part of the Kyoto Encyclopedia of Genes and Genomes (KEGG) Huntington’s disease gene pathway but is also directly involved in aberrant BDNF transport [45]. *DNAL1* is one of three genes associated with dynein axonemal light chain 4, part of the ATP-dependent outer dynein arms complex that acts as a molecular motor for cilia [31].

The final intronic SNP, rs7655305, regulates the expression of *FAM114A1* (mostly in the medulla, substantia nigra, and hippocampus) [40] and other genes, especially in the cerebellum (Table 3). This variant is an eQTL for *TLR1* in the DLPFC (transcript and exon levels in the disease population) [39] as well as other genes in many areas outside the brain, including skin (especially sun-exposed), tibial nerve, subcutaneous and visceral adipose, breast mammary tissue, testis, thyroid, and esophagus. rs7655305 is associated with enhancer histone marks in four tissues outside of the brain, the alteration of CCCTC-binding factor (CTCF) and telomere length regulation (Tel2) regulatory motifs [41], and active H3K36me3 (transcription-associated) histone modification in the middle hippocampus (http://www.featSNP.org). This SNP is also in LD with several other intronic *FAM114A1* SNPs with correlated enhancer histone marks, altered regulatory motifs, and eQTL hits [41].

rs112140519 is a deletion/insertion variation closest to *NUS1* that is linked to H3K4me3 promotion in adipose nuclei and duodenum smooth muscle as well as the alteration of five regulatory motifs (Dbx1, Hoxb13, Ncx, ZNF263, Zfp105) [41]. rs548321, an intergenic SNP 70 kbps from the 5′ end of *LRRC55*, most strongly affects the expression of *TIMM10*, *UBE2L6*, and *ZDHHC5* (especially in the putamen, although *TIMM10* effects were most prominent in the frontal cortex). Each of these genes exhibits the lowest expression in the cerebellum relative to other brain tissue types [40]. Haploreg data shows selective association with H3K4me1 and H3K27ac enhancements in anterior caudate, as well as the alteration of GR and Pax-4 regulatory motifs [41]. rs427790, an intergenic SNP between *MIR181A1* and *NR5A2*, is an eQTL hit in the testis and basal ganglia [43]. Haploreg indicates the associated alteration of double homeodomain (Dux1), Foxj1, and Zec regulatory motifs [41].

### 4.6. Influence of HTT CAG-Repeats

Including the CAG-repeat number or CAG-age-product (CAP) score as a covariate did not affect the significance of the GMC-SNP component correlation. Similarly, the inclusion of the CAP did not nullify the significance of the NTRK2-SNP correlation with TMTB, although it did account for some of the variance. While we can currently only speculate on this finding, it may be attributable to the regression of age effects from the imaging data prior to pICA. The CAP reflects the cumulative disease burden of age and the CAG-repeat number. Because the GMC component had the effects of age subtracted from it, CAP and CAG would be expected to display a similar relationship to the GMC component, which is what we observed. The relationships between the GMC component and the clinical variables, however, could be accounted for by the CAP, likely reflecting the sensitivity of these measures to CAG and age, both of which are highly related to the clinical variables assessed in this study.

## 5. Conclusions

The study results demonstrated the following:In this PREDICT-HD prodromal cohort, high or low levels of an SNP profile with substantial contributions from *NTRK2* were associated with a GMC profile representing the supplementary and primary motor cortex, as well as other frontal regions (positive correlation).This frontal gray matter profile was associated with cognitive and motor performance in this population.The SNP component was not significantly associated with clinical functioning, but one of its top *NTRK2* SNPs had a protective association with performance on TMTB, a measure of task switching and visual attention, indicating some influence on cognition.Correlations between the SNP component and clinical/GMC variables were mainly due to top contributing SNPs, rather than being an aggregate effect of the entire SNP component.Top component SNPs have been associated with active histone modifications in the brain (cingulate gyrus, inferior temporal, angular gyrus, DLPFC, caudate, hippocampus, and substantia nigra) and altered regulatory motifs (especially the glucocorticoid receptor (GR) and zinc finger protein (Zec)).Top *NTRK2* SNPs in the component were close to alternative stop codons and reportedly regulated genes implicated in diverse functions (especially in the frontal cortex, thalamus, putamen, and cerebellum).

Although further investigation is warranted, these results suggest that *NTRK2* has protective potential in Huntington’s disease, especially in individuals with certain genotypes. Treatments that target BDNF receptors may help preserve frontal gray matter that is important for cognitive and motor functioning.

## 6. Limitations

In any study involving many participants with a rare condition, multiple scanning sites are typical and a certain degree of inhomogeneity in data collection is thus inevitable. This study incorporates data from several unique 1.5 T and 3 T MRI scanners, and every effort was made to control for possible confounds related to this. Before data collection, uniform protocols were established to ensure maximal homogeneity of data collection. The effects of collection site and scanner field strength, as well as sex and age, were regressed from GMC images before pICA.

Ideally, controls would be included in the analysis to permit comparisons with prodromal HD participants. PICA was carried out with a more extensive dataset (*N* = 903) that included controls (*N* = 189). However, the analysis did not extract a significantly correlated SNP-GMC component pair that withstood correction for multiple testing. The effects presented in this study may be unapparent in the combined sample with controls because they are related to other processes already aberrant in HD, such as BDNF transcription and nuclear translocation.

Because the GMC-SNP component correlation was driven by a small percentage of participants, the results may not apply to the entire HD population, and treatments that target *NTRK2* should consider the context of the patient’s genomic landscape. Nonetheless, for cancer and other diseases, emphasis is shifting away from blanket treatments toward more personalized gene therapies that consider individual genotypes. If clinical trials to reduce mHTT are ineffective or harmful for certain individuals in the long-run, therapies that instead target mediators of mHTT toxicity may provide a promising alternative.

## 7. Future Directions

This study helped characterize the complex interactions of BDNF-signaling genes, brain structure, and clinical functioning in prodromal HD. As genetic and epigenetic factors of prodromal HD progression continue to be identified, potential gene therapies (such as DNA-methylating drugs and histone-deacetylase inhibitors) can be refined, tested, and implemented more strategically [46]. To aid the development of targeted treatments, future studies should test the validity of strong correlations observed in the most extensive available human datasets by establishing causal links in non-human animals. For example, many HD *Drosophila* (fruit fly) strains are available that permit rapid, efficient, and inexpensive querying of SNPs and genes of interest derived from human studies. The most promising candidates from these experiments can be flagged for continued research in mammals and eventual use in clinical trials. A promising direction is to examine how different ratios of BDNF TrkB and p75 receptors influence progression and onset in the context of *HTT* CAG-repeats. BDNF acts on TrkB receptors to increase its own continued expression; thus, an underrepresentation of these receptors may reduce BDNF. Similarly, p75 receptor overrepresentation or an overabundance of pro-BDNF may promote apoptosis and contribute to disease-related decline. Another important consideration is the impact of other BDNF exons. Huntingtin stimulates transcription from the BDNF exon II promoter, which is 60% less active in cells overexpressing mHTT [9]. Although BDNF exon III and IV promoter actions are not linked to huntingtin, mHTT reduces their transcription through unknown mechanisms thought to involve cAMP Response Element Binding protein (CREB) and CREB-Binding Protein (CBP). Thus, promoter II transcriptional reduction may be linked to huntingtin reductions, while that of promoters III and IV may be related to mHTT toxicity. CREB and CBP are also linked to several altered functions in HD, and their investigation is a valuable avenue for future research. This study examined BDNF-signaling genes in relation to GMC and clinical functioning. However, white matter is dramatically affected in HD and may also be influenced by these factors, a possibility that should be examined in future studies. Aside from genetic influences on HD progression, epigenetic marks (which can alter gene expression) also differ in patients compared to controls. This may be an especially promising therapeutic direction because, unlike the fixed DNA sequence, the epigenome changes across the lifespan and in response to environmental stimuli. Future studies should seek to characterize prodromal epigenetic differences in various tissues, as well as identify protective and detrimental variations in pathways with known clinical relevance to HD. An additional important future direction is to characterize these factors longitudinally. SNP and methylation patterns, for example, in BDNF/REST pathways may be associated with different rates of clinical decline or the rapidity of HD onset. Factors associated with disease resilience could then be incorporated into intervention strategies to slow or prevent HD conversion.

## Figures and Tables

**Figure 1 brainsci-08-00116-f001:**
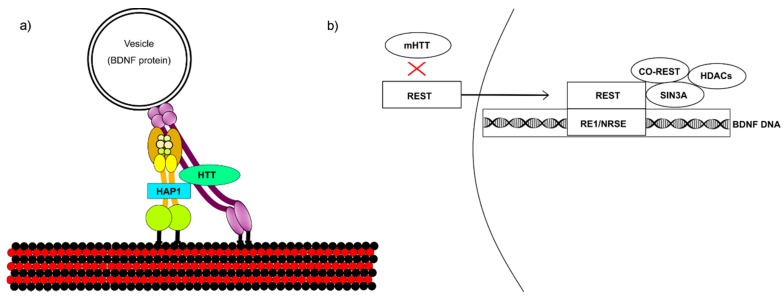
Mutant huntingtin (mHTT) affects (**a**) brain-derived neurotrophic factor (BDNF) vesicular transport across microtubules and (**b**) BDNF transcription. (**a**) Huntingtin binds to HAP1 to recruit vesicular transport proteins. The dynein complex is shown in yellow and the dynactin complex in purple. Together, these proteins enable retrograde vesicular transport, which is disrupted by mHTT binding too tightly to HAP1. (**b**) Huntingtin sequesters RE1 silencing transcription factor (REST) in the cytoplasm, inhibiting its accumulation in the nucleus. If mHTT fails to sequester REST, this leads to its nuclear buildup and consequently reduces BDNF transcription.

**Figure 2 brainsci-08-00116-f002:**
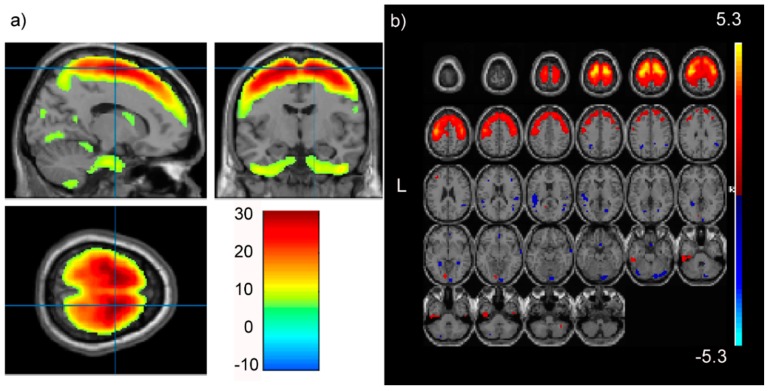
The frontal gray matter concentration (GMC) component significantly paired with the *NTRK2* SNP component in pICAr: (**a**) maximum effects in right premotor and supplementary motor. Crosshairs are positioned at the global maximum (T(1) = 30.75) and reach a threshold at *p* = 0.05; (**b**) multi-slice axial topography (threshold: *Z* = 2.5) showing a mostly positive component with strong representation from superior frontal gray matter and Brodmann area 6 (supplementary and premotor cortex). Images are displayed using xjView (a; http://www.alivelearn.net/xjview) and Fusion ICA Toolbox (b; http://mialab.mrn.org/software/fit).

**Figure 3 brainsci-08-00116-f003:**
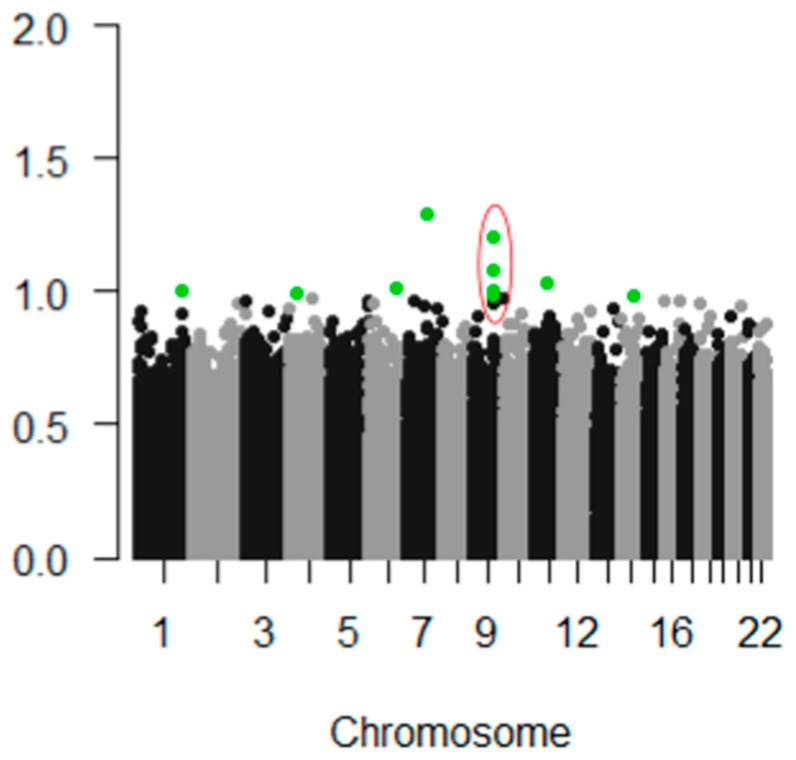
Manhattan plot showing top-weighted SNPs within the SNP component that correlated with a frontal/supplementary motor gray matter profile in parallel ICA. The top 10 SNPs contributing to the component are highlighted in green, and the four top *NTRK2* SNPs are circled in red. The *Y* axis indicates each SNP’s weight, or contribution, to the SNP component. The plot was generated using the qqman package in R version 3.4.1.

**Figure 4 brainsci-08-00116-f004:**
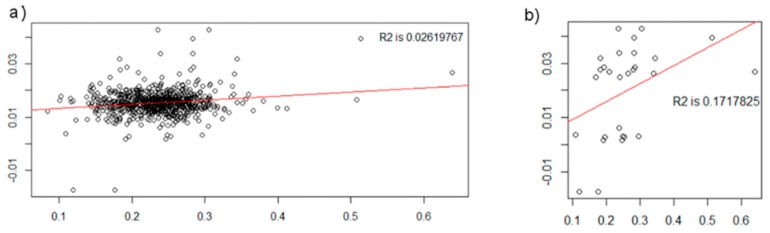
Correlation between pICA frontal GMC and *NTRK2* SNP components in (**a**) the full prodromal sample and (**b**) participants with SNP component values ≥2 SDs above or below the mean (*N* = 28). Mean SNP component value = 0.015 (full prodromal sample), SD = 0.005, *X* axis = SNP component weights, *Y* axis = GMC component weights, R2 = *r*^2^. Graphics generated with R version 3.4.1.

**Figure 5 brainsci-08-00116-f005:**
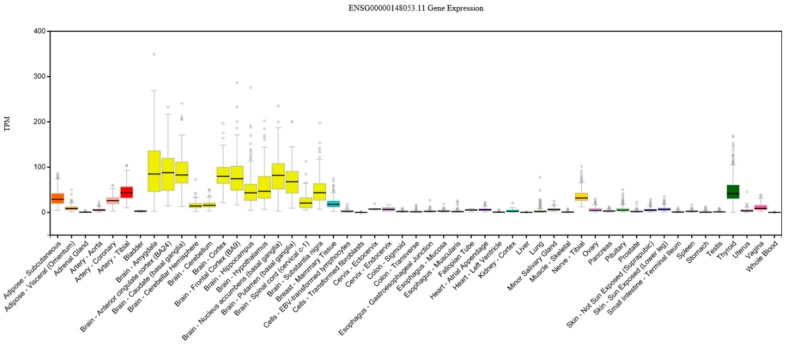
Expression of *NTRK2* in various tissue types, showing relatively strong expression in brain tissue. TPM = transcripts per kilobase million. Expression threshold: >0.1 TPM and ≥6 reads in 20% or more of samples. Box plots are 25th and 85th percentiles, with a black line at the median. Outliers are ±1.5 times the interquartile range. Data source: GTEx Analysis Release V7 (dbGaP Accession phs000424.v7.p2).

**Table 1 brainsci-08-00116-t001:** Genes from which available single nucleotide polymorphisms (SNPs) were included as references for parallel independent component analysis with reference (pICAr).

Factor	Gene(s)	pICA Reference SNPs (#)	Full Name	Function
REST/NRSF	*REST*, *RCOR1*, *RCOR3* ^†^	4	RE1 silencing transcription factor/neuron-restrictive silencer factor	Transcriptional repression
Sin3A	*SIN3A*	2	SIN3 transcription regulator family member A	Part of co-repressor complex with REST and coREST
CoREST	*RCOR1*	9	REST co-repressor	Part of co-repressor complex with REST and Sin3A
HAP1	*HAP1*	7	Huntingtin-associated protein 1	Binds to huntingtin, facilitates brain-derived neurotrophic factor (BDNF) transcription and transport
TrkB	*NTRK2*	52	Tropomyosin receptor kinase B	BDNF high-affinity receptor
P75	*NGFR*	26	Low-affinity nerve growth factor receptor	BDNF low-affinity receptor
RILP	*RILP*	5	REST-interacting LIM domain protein	REST nuclear receptor
Sortilin	*SORT1*	12	Sortilin 1	Suggested apoptotic function with p75 and pro-BDNF
BDNF	*BDNF*	6	Brain-derived neurotrophic factor	Neuronal growth, survival, differentiation

^†^ SNPs from this gene were not available. # = number.

**Table 2 brainsci-08-00116-t002:** The 10 SNPs most heavily weighted in the SNP component, with reference SNP cluster IDs (rs IDs), weights in the component and their directions, associated genes, minor allele frequencies, and variant class. SNV = single nucleotide variation, DIV = deletion/insertion variation.

SNP	Weight (||)	+/−	Ranking	Gene	Minor Allele Frequency	Class
rs7801922	1.29	+	1	*CDK14*	T = 0.34/1704 (1000 Genomes) T = 0.38/11000 (TOPMED)	SNV
rs11140810	1.20	+	2	*NTRK2*	G = 0.42/2104 (1000 Genomes) G = 0.42/12329 (TOPMED)	SNV
rs4877289	1.08	+	3	*NTRK2*	G = 0.38/1926 (1000 Genomes) G = 0.38/11160 (TOPMED)	SNV
rs548321	1.03	+	4	70 kb 5′ of *LRRC55*	G = 0.41/2055 (1000 Genomes) G = 0.38/11140 (TOPMED)	SNV
rs112140519	1.01	+	5	53 kb 3′ of *NUS1*	-=0.33/1652 (1000 Genomes)	DIV
rs427790	1.00	+	6	*MIR181A1*, *NR5A2*	C = 0.33/1658 (1000 Genomes) C = 0.38/10948 (TOPMED)	SNV
rs10868241	1.0	+	7	*NTRK2*	A = 0.32/1614 (1000 Genomes) A = 0.24/6986 (TOPMED)	SNV
rs7655305	0.99	+	8	*FAM114A1*	G = 0.43/2140 (1000 Genomes) G = 0.43/12519 (TOPMED)	SNV
rs2277193	0.98	+	9	*NTRK2*	C = 0.34/1679 (1000 Genomes) C = 0.41/11827 (TOPMED)	SNV
rs8012614	0.98	+	10	*HEATR4*	C = 0.29/1442 (1000 Genomes) C = 0.37/10860 (TOPMED)	SNV

**Table 3 brainsci-08-00116-t003:** Top SNPs contributing to supplementary/frontal-related pICA SNP components and reported effects on gene expression in the brain. These SNPs are associated with gene expression changes most prominently in the frontal lobe (in agreement with the study results), thalamus, putamen, and cerebellum. The first column lists the SNP rs ID, associated or closest gene, and gene type (I = intronic, IG = intergenic). Occ. = occipital, Thal. = thalamus, Temp. = temporal, WM = white matter, Put = putamen, Hipp. = hippocampus, Fron. = frontal, Cereb. = cerebellum, SNigra = substantia nigra, Med. = medulla, DLPFC = dorsolateral prefrontal cortex. ^†^ = Genes associated with more than one SNP and more than tissue type; ^*^ = Genes associated with more than one tissue type. Expression data was obtained from Braineac [40].

SNP (rs)	Occ.	Thal.	Temp.	WM	Put.	Hipp.	Fron.	Cereb.	SNigra	Med.	DLPFC
8012614 *HEATR4* (I)	*NUMB*; *TMEM90A*	*HEATR4* ^*^			*ACOT4* ^*^	*RBM25* ^*^; *ACOT4* ^*^		*RBM25* ^*^	*RBM25* ^*^		*RBM25* ^*^; *ACOT4* ^*^; *HEATR4* ^*^; *DNAL1*
7801922 *CDK14* (I)	*STEAP1*	*C7orf63; STEAP2* ^*^		*STEAP2* ^*^		*C7orf63* ^*^		*STEAP2* ^*^	*DPY19L2P4; CDK14* ^*^		*C7orf63* ^*^
7655305 *FAM114A1* (I)			*RPL6*	*PDS5A*		*FAM114A1* ^*^	*TLR6*	*PTTG2; FLJ13197; UGDH*	*FAM114A1* ^*^	*FAM114A1* ^*^	*TLR1*
548321 *LRRC55* (IG)					*UBE2L6; ZDHHC5*		*TIMM10*				
427790 *MIR181A1, NR5A2* (IG)		*NEK7*	*NR5A2* ^*^; *PTPRC; ATP6V1G3*		*NR5A2* ^*^; *MIR181A1* ^*^		*MIR181A1* ^*^	*MIR181A1* ^*^	*NR5A2* ^*^		
11140810 *NTRK2* (I)	*NTRK2* ^†^	*NTRK2* ^†^; *HNRNPK* ^†^; *MIR7-1* ^†^		*NTRK2* ^†^; *SLC28A3* ^†^	*NAA35* ^†^	*NTRK2* ^†^	*NTRK2* ^†^	*HNRNPK* ^†^; *MIR7-1* ^†^		*SLC28A3* ^†^	*NTRK2* ^†^
2277193 *NTRK2* (I)		*SLC28A3* ^†^; *HNRNPK* ^†^; *MIR7-1* ^†^		*SLC28A3* ^†^		*NAA35* ^†^	*SLC28A3* ^†^; *AGTPBP1* ^†^	*SLC28A3* ^†^; *HNRNPK* ^†^; *MIR7-1* ^†^; *NAA35* ^†^; *AGTPBP1* ^†^	*NAA35* ^†^	*SLC28A3* ^†^	
4877289 *NTRK2* (I)	*AGTPBP1* ^†^			*HNRNPK* ^†^; *MIR7-1* ^†^	*SLC28A3* ^†^		*RMI1* ^†^; *SLC28A3* ^†^; *AGTPBP1* ^†^				
10868241 *NTRK2* (I)		*HNRNPK* ^†^; *MIR7-1* ^†^		*SLC28A3* ^†^	*RMI1* ^†^	*SLC28A3* ^†^	*SLC28A3* ^†^; *NTRK2* ^†^			*HNRNPK* ^†^; *MIR7-1* ^†^; *AGTPBP1* ^†^	*NTRK2* ^†^

**Table 4 brainsci-08-00116-t004:** Genes regulated by prominent *NTRK2* SNPs in this study, along with their associated functions and pathways. Sources: GeneCards human gene database [31] and UniProt [42].

Gene Name	Full Name	Associated *NTRK2* SNP(s)	Description	Type	Related Pathways
*SLC28A3*	Solute Carrier Family 28 Member 3	rs1114081, rs2277193, rs4877289, rs10868241	Neurotransmission, vascular tone, adenosine concentration near cell surface receptors, transport/metabolism of nucleoside drugs	Protein coding, nucleoside transporter	Vitamin and nucleoside transport, thiopurine pathway, pharmacokinetics/pharmacodynamics
*AGTPBP1*	ATP/GTP Binding Protein 1	rs10868241	Contains nuclear localization signals and an ATP/GTP-binding motif, involved in the deglutamylation of protein polyglutamate side chains, removal of gene-encoded polyglutamates from protein carboxy-terminus, and shortening of long polyglutamate chains	Protein coding, zinc carboxypeptidase, metallocarboxypeptidase	Neuroscience
*HNRNPK*	Heterogeneous Nuclear Ribonucleoprotein K	rs1114081, rs2277193, rs4877289, rs10868241	Major pre-mRNA-binding protein, binds to poly(C) sequences, involved in nuclear metabolism of hnRNAs, and p53/TP53 response to DNA damage (transcriptional activation and repression)	Protein coding, heterogeneous nuclear ribonucleoprotein (hnRNP)	Translational control and mRNA splicing
*MIR7-1*	MicroRNA 7-1	rs1114081, rs2277193, rs4877289, rs10868241	Affiliated with an undefined RNA class	RNA gene	mRNA splicing, SUMOylation
*NAA35*	N(Alpha)-Acetyltransferase 35, NatC Auxiliary Subunit	rs1114081, rs2277193	Involved in the regulation of apoptosis, and the proliferation of smooth muscle cells	Protein coding	Golgi-to-endoplasmic reticulum (ER), trans-Golgi-network retrograde transport

**Table 5 brainsci-08-00116-t005:** Histone modifications contributing to chromatin states in the top *NTRK2* SNPs in the brain. Altered regulatory motifs associated with SNPs are also listed. H3K27ac = enhancer/promoter-associated, H3K4me1 = enhancer-associated. CG = cingulate gyrus, IT = inferior temporal, AG = angular gyrus, DLPFC = dorsolateral prefrontal cortex, Ant. Caud. = anterior caudate, MHipp = middle hippocampus, SNigra = substantia nigra. Data from HaploReg v4.1 [41].

*NTRK2* SNP	CG	IT	AG	DLPFC	Ant. Caud.	MHipp	SNigra	Regulatory Motifs Altered
rs11140810	H3K27ac	H3K27ac	H3K27ac	H3K27ac				Foxa, GLI, Hic1, Zec
rs4877289	H3K4me1				H3K4me1			
rs10868241	H3K27ac, H3K4me1	H3K27ac, H3K4me1	H3K27ac	H3K27ac, H3K4me1	H3K27ac	H3K27ac, H3K4me1	H3K27ac, H3K4me1	
rs2277193	H3K27ac	H3K27ac	H3K27ac	H3K27ac				GR, Pax-6

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
