# Peer review of "High and Low Levels of an NTRK2-Driven Genetic Profile Affect Motor- and Cognition-Associated Frontal Gray Matter in Prodromal Huntington’s Disease"

_brainsci, 2018, doi:10.3390/brainsci8070116_

Round 1

Reviewer 1 Report

The paper is an original research articles describing well the hypothesis, study, and methods.

I think that the paper is well written and that the matter of the study deserve scientific attention.

The methodology used seem to be sound enough to test the hypothesis.

 However, I have to highlight some issues that authors should address before considering the paper publishable. 

-          I think that authors should focus their attention on cognitive performance (eg drawing a little table). 

-          Row 171: the Authors means age 18-82 for the entire group or only for females ?

-          The gene that you mentioned on row 331 is FAM11RA1 is not the same you showed in table 2 FAM114A1 ?

Author Response

Response to Reviewer 3 Comments

Point 1: I think that authors should focus their attention on cognitive performance (eg drawing a little table). 

Response 1: Thank you very much for your time and feedback. As part of our response to reviewer 1, we have included additional details regarding the clinical variables chosen and clarified our reasoning for choosing them (lines 195-207 of the cognitive and motor variables section):

“Seven clinical variables were selected and tested for brain structural and genetic associations (outlined below), based on their established clinical reliability and sensitivity to prodromal HD progression [16]. Because we were interested in genetic and brain-structural effects on cognitive and motor functioning, we analyzed the portions of the Unified Huntington Disease Rating Scale (UHDRS) relevant to these domains. The UHDRS includes four sections measuring movement, cognition, behavior, and functional capacity. We assessed the Total Motor Score (UHDRS-TMS), which comprises the movement portion of the UHDRS and is a sum of the scores on the individual motor variables (oculo-motor function, dysarthria, chorea, dystonia, gait, and postural stability). We also analyzed the Symbol Digit Modalities Test (SDMT) and the Stroop test (Color, Word, and Interference conditions), which are two of the three parts of the UHDRS cognition section (we did not assess verbal fluency). In addition to the UHDRS-TMS, SDMT, and Stroop Interference, we also analyzed Trail-Making Tests A and B (TMTA and TMTB), which are not part of the UHDRS. We chose these tests because of our specific interest in cognition and motor functioning, and because previous work by the PREDICT-HD group has demonstrated that these measures are particularly sensitive to prodromal changes in brain structure [16].”

We have also clarified the distinction between Stroop Interference and the other Stroop conditions (lines 233-234), and added a sentence to the discussion (lines 470-480) acknowledging the lack of significant effects for SDMT and Stroop Word and offering a possible interpretation:

“This may also explain why significant effects were not observed for SDMT and Stroop Word, as motor involvement in these tasks is relatively limited.”

Point 2: Row 171: the Authors means age 18-82 for the entire group or only for females?

Response 2: The provided age range pertains to the entire sample. In the text, we have adjusted the presentation of this information to make this clearer, as follows:

A cohort of 715 gene positive (≥36 CAG-repeats) prodromal PREDICT-HD participants were included in the study [448 females and 267 males: ages 18-82, mean CAG-repeat number = 42.5 (SD = 2.5)].”

Point 3: The gene that you mentioned on row 331 is FAM11RA1 is not the same you showed in table 2 FAM114A1?

Response 3: The correct gene name is FAM114A1; the manuscript has been updated to reflect this.

Reviewer 2 Report

This is an excellent paper, well-written and comprehensive. The authors do well to highlight limitations without significant controls, and better harmonization of imaging across sites in future studies will be essential. Why do the authors think CAG or CAP covariate analysis did not influence the GMC-SNP component relationship or NTRK2/TMTB relationship? A comment on this seems worthwhile. 

Author Response

Response to Reviewer 2 Comments

Point 1: This is an excellent paper, well-written and comprehensive. The authors do well to highlight limitations without significant controls, and better harmonization of imaging across sites in future studies will be essential. Why do the authors think CAG or CAP covariate analysis did not influence the GMC-SNP component relationship or NTRK2/TMTB relationship? A comment on this seems worthwhile.

Response 1: We appreciate the reviewer’s comments and feedback. We have updated the “Influence of HTT CAG-repeats” section (lines 624-634) to better explain our interpretation of these findings:

“Including CAG-repeat number or CAG-age-product (CAP) score as a covariate did not affect the significance of the GMC-SNP component correlation. Similarly, inclusion of CAP did not nullify the significance of the NTRK2-SNP correlation with TMTB, although it did account for some of the variance. While we can currently only speculate on this finding, it may be attributable to the regression of age effects from the imaging data prior to pICA. CAP reflects the cumulative disease burden of age and CAG-repeat number. Because the GMC component had the effects of age subtracted from it, CAP and CAG would be expected to display a similar relationship to the GMC component, which is what we observed. The relationships between the GMC component and the clinical variables, however, could be accounted for by CAP, likely reflecting the sensitivity of these measures to CAG and age, both of which are highly related to the clinical variables assessed in this study.”

Reviewer 3 Report

My only comment is that the authors spoke about the Total Motor Score Stroop and Trials A&B.  I did not see results for the rest of the UHDRS. Perhaps other reviewers will comment on this

Author Response

Response to Reviewer 1 Comments

Point 1: My only comment is that the authors spoke about the Total Motor Score Stroop and Trials A&B.  I did not see results for the rest of the UHDRS. Perhaps other reviewers will comment on this

Response 1: We appreciate your feedback and agree that it should be clearer which portions of the UHDRS were analysed and why. Additionally, there should be some commentary regarding which variables had significant associations and which did not.

To address this, the cognitive and motor variables section (lines 195-207) has been modified to clarify the structure of the UHDRS and explain which portions were included in analyses as well as why these variables were chosen:

“Seven clinical variables were selected and tested for brain structural and genetic associations (outlined below), based on their established clinical reliability and sensitivity to prodromal HD progression [16]. Because we were interested in genetic and brain-structural effects on cognitive and motor functioning, we analyzed the portions of the Unified Huntington Disease Rating Scale (UHDRS) relevant to these domains. The UHDRS includes four sections measuring movement, cognition, behavior, and functional capacity. We assessed the Total Motor Score (UHDRS-TMS), which comprises the movement portion of the UHDRS and is a sum of the scores on the individual motor variables (oculo-motor function, dysarthria, chorea, dystonia, gait, and postural stability). We also analyzed the Symbol Digit Modalities Test (SDMT) and the Stroop test (Color, Word, and Interference conditions), which are two of the three parts of the UHDRS cognition section (we did not assess verbal fluency). In addition to the UHDRS-TMS, SDMT, and Stroop Interference, we also analyzed Trail-Making Tests A and B (TMTA and TMTB), which are not part of the UHDRS. We chose these tests because of our specific interest in cognition and motor functioning, and because previous work by the PREDICT-HD group has demonstrated that these measures are particularly sensitive to prodromal changes in brain structure [16].”

Additionally, a sentence was added to clarify the distinction between Stroop Interference and the other Stroop conditions (lines 233-234):

The interference condition measures the ability to inhibit the dominant (or automatic processing) response, which is to read the word.”

An additional sentence was added to the discussion at lines 470-480 acknowledging the lack of significant effects for SDMT and Stroop Word and offering a possible interpretation:

“This may also explain why significant effects were not observed for SDMT and Stroop Word, as motor involvement in these tasks is relatively limited.”